# Antibody-controlled actuation of DNA-based molecular circuits

Wouter Engelen[1], Lenny H.H. Meijer[1], Bram Somers[1], Tom F.A. de Greef[1] & Maarten Merkx[1]

DNA-based molecular circuits allow autonomous signal processing, but their actuation has relied mostly on RNA/DNA-based inputs, limiting their application in synthetic biology, biomedicine and molecular diagnostics. Here we introduce a generic method to translate the presence of an antibody into a unique DNA strand, enabling the use of antibodies as specific inputs for DNA-based molecular computing. Our approach, antibody-templated strand exchange (ATSE), uses the characteristic bivalent architecture of antibodies to promote DNA-strand exchange reactions both thermodynamically and kinetically. Detailed characterization of the ATSE reaction allowed the establishment of a comprehensive model that describes the kinetics and thermodynamics of ATSE as a function of toehold length, antibody–epitope affinity and concentration. ATSE enables the introduction of complex signal processing in antibody-based diagnostics, as demonstrated here by constructing molecular circuits for multiplex antibody detection, integration of multiple antibody inputs using logic gates and actuation of enzymes and DNAzymes for signal amplification.

[1] Laboratory of Chemical Biology and Institute of Complex Molecular Systems, Department of Biomedical Engineering, Eindhoven University of Technology, PO Box 513, 5600 MB Eindhoven, The Netherlands. Correspondence and requests for materials should be addressed to M.M. (email: m.merkx@tue.nl).

Governed by its highly automated synthesis and the predictable and modular nature of self-assembly, DNA has emerged as a versatile building block for the construction of precisely defined nanometer-sized structures and sophisticated molecular circuits[1–7]. Examples range from three-dimensional nanostructures capable of encapsulating therapeutic molecules to complex DNA-based molecular circuits, robots and motors[8–11]. DNA-based molecular circuits capable of adaptation, oscillations and bistability have been created using enzymes involved in DNA synthesis and degradation such as DNA polymerases, nickases and exonucleases[12–14]. Moreover, toehold-mediated strand exchange has been introduced as a highly versatile and universal molecular programming language to construct enzyme-free systems that include logic gates, signal amplification, thresholding, feedback control and consensus gating[15,16]. In toehold-mediated strand exchange reactions, complementary single-stranded domains (toeholds) allow two DNA reactants to transiently hybridize, thereby initiating dynamic strand exchange reactions. By tuning the length and sequence of a toehold, both the kinetics and thermodynamics of toehold-mediated exchange reactions can be readily controlled[17–19]. Impressive examples of DNA-based molecular circuits based on this principle include neural networks[20] and adaptive immune system mimics[21], molecular tic-tac-toe[22] and circuits able to calculate the square root of four bit binary input numbers[6].

Whereas oligonucleotide-based molecular circuits are not expected to rival silicon-based electronic computing, their key advantage is that they can be integrated at a molecular level with biological systems[23–26]. For example, autonomous DNA-based molecular circuits that are able to sense specific input signals in their environment, process these inputs according to a predefined algorithm, and finally translate the result into a biological activity could be used as theranostic devices[27]. The successful application of DNA-based molecular circuits for these and other applications in bottom up synthetic biology relies on their ability to sense and act on their environment. While some progress has been made to control downstream protein activity using DNA-based molecular circuits, the upstream actuation of DNA-based molecular circuits still mostly relies on oligonucleotide-based input triggers[28–30]. With the exception of a few well-characterized protein-binding aptamers, generic design principles to interface DNA-based molecular circuits with protein-based input triggers are lacking[31].

The excellent specificity and affinity of antibody-based molecular recognition has proven invaluable for the development of modern diagnostic assays and therapeutic antibodies constitute an important class of newly introduced drugs[32]. In addition, antibodies represent excellent biomarkers for a range of diseases, in particular infectious and autoimmune diseases. Their omnipresence in today's lifesciences encouraged us to develop a generic approach to use antibodies as specific inputs for DNA-based molecular computing. Our strategy harnesses the characteristic bivalent Y-shaped molecular architecture of antibodies as a template to promote strand exchange of peptide-functionalized DNA strands, providing a generic and highly efficient way to translate the presence of an antibody into a specific DNA output sequence. In this work, we present a detailed experimental characterization and optimization of this antibody-templated strand exchange (ATSE) reaction, introduce a model to understand the critical parameters that determine its kinetic and thermodynamic properties, and demonstrate how DNA-based molecular circuits can be used to process multiple antibody inputs using predetermined logic operations and control downstream catalytic systems.

## Results

**Antibody-templated strand exchange reactions.** Figure 1a shows the principle of the antibody-templated toehold-mediated strand exchange reaction. The ATSE system consists of a base strand (**B**) and an output strand (**O**) prehybridized to form duplex **BO**, and an invading strand (**I**). A toehold (*T*) on **BO** allows **I** to bind and displace **O**, but the number of basepairs in **BO** is higher than in **BI**. In the absence of the target antibody this reaction is therefore thermodynamically unfavourable and is maintained in the initial state, that is, no output is generated. Conjugation of antibody-specific peptide epitopes to the 3′-ends of **B** and **I**, allows binding of **BO** and **I** to their target antibody and enhances the toehold exchange reaction in two ways. First, the product of the exchange reaction **BI** can form a bivalent interaction with its target antibody, thus making the displacement of **O** by **I** thermodynamically more favourable. We recently showed that bivalent peptide-dsDNA ligands form very tight 1:1 complexes with their target antibody, showing a 500-fold increase in affinity compared to the monovalent peptide-antibody interaction[33,34]. Second, the colocalization of the reactants increases their effective concentration, hence enhancing the rate of the exchange reaction.

To provide proof-of-principle for ATSE we used a monoclonal antibody targeting the HA-tag, a peptide derived from the human influenza virus hemagglutinin protein ($K_d = 0.78$ nM, Supplementary Fig. 9). The kinetics of the strand exchange reaction was studied by coupling the released output strand **O** to a downstream toehold-mediated strand displacement (TMSD) reaction with a reporter duplex (**Rep**), resulting in quantitative release of a fluorescently labelled oligonucleotide. To ensure minimal background in combination with fast initiation of the ATSE reaction, we optimized the design by systematically increasing the length of the toehold *T* from 0 to 6 nucleotides, hence gradually destabilizing the initial state. The length of the strands was chosen such that the bivalent peptide-dsDNA product **BI** contained 24 basepairs, a length that we have previously shown to be optimal for bivalent binding to the target antibody[33,34]. In the absence of input antibody no increase in fluorescence was observed up to a toehold length of 3 nucleotides (Fig. 1b). Increasing the toehold beyond 3 nt gradually increased the rate of the background reaction until it was maximal at a toehold length of 6 nucleotides. In the presence of 5 nM of anti-HA antibody, a significant amount of strand exchange was observed even for a 1 nucleotides toehold and the maximum rate was already obtained at a toehold length of 3 nucleotides. Figure 1c shows the apparent first-order rate constants for the ATSE reaction as a function of toehold length in the presence and absence of anti-HA antibody. Dividing the apparent rate constant in the presence of antibody by the apparent rate constant of the background reaction shows that a maximal signal-to-background ratio (S/B ratio) of ∼100 was obtained using a toehold length of 3 nucleotides (Fig. 1d). The limit of detection (LOD) was determined to be 23 pM under these conditions (Supplementary Fig. 5).

The modularity of the ATSE principle should allow its application to virtually any antibody by simply exchanging the peptide epitopes. To test this, we repeated these experiments for the anti-HIV1-p17 antibody, using peptide epitopes derived from the p17 coat protein of the HIV1 virus that bind with a monovalent $K_d$ of 16 nM (Supplementary Fig. 10). At the same concentration of antibody (5 nM), the rate of ATSE was found to be smaller for the anti-HIV1-p17 antibody compared to the anti-HA-antibody, which can be explained by the lower affinity of the peptide-antibody interaction in the former, meaning that less reactants will form the reactive peptide-antibody complex. However, despite the significantly lower binding affinity of the HIV1 epitope to the antibody, the ATSE reaction displayed a similar toehold length dependence as the anti-HA antibody

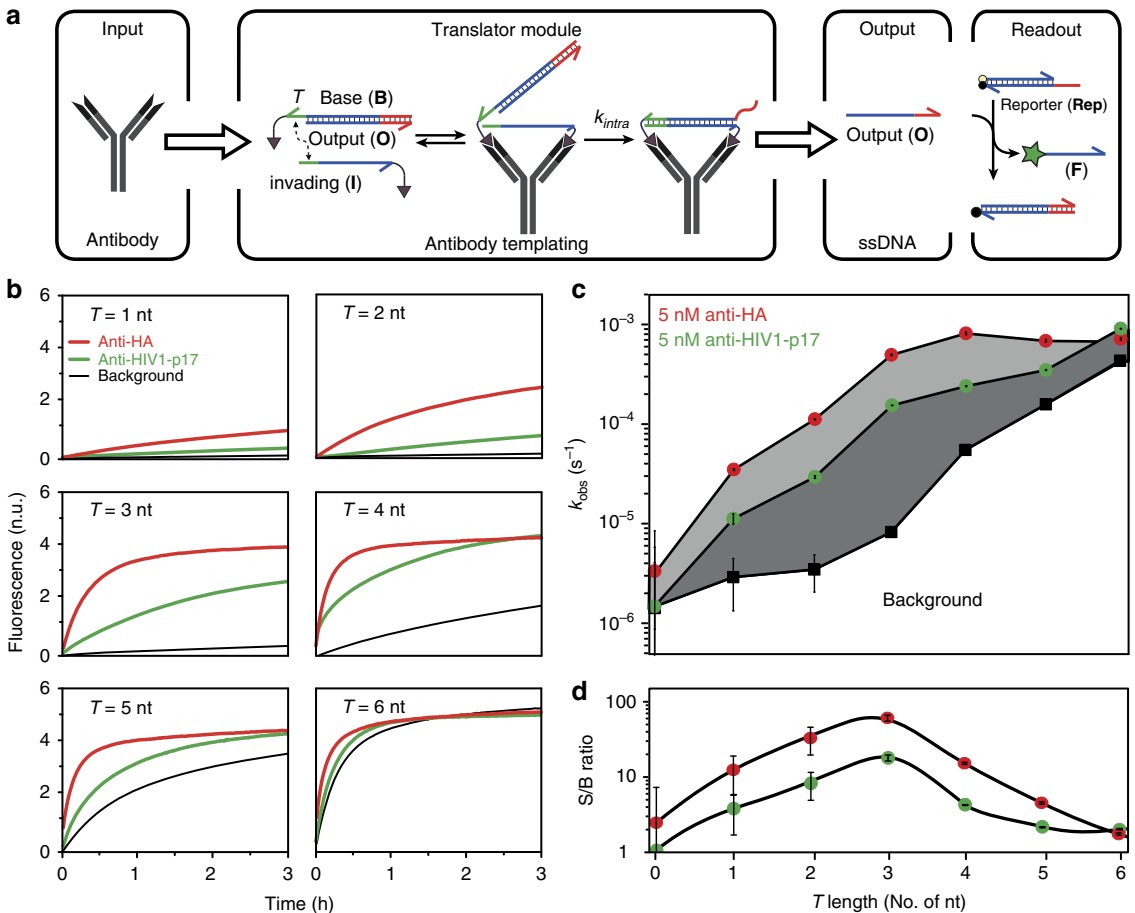

**Figure 1 | Antibody-templated strand exchange allows translation from antibody to DNA.** (**a**) Principle of ATSE. In the absence of antibody (**Ab**) the strand exchange reaction is thermodynamically unfavourable and remains in the initial state with output strand (**O**) hybridized to the peptide-functionalized base strand (**B**). Scaffolding of the oligonucleotide reactants on the input antibody initiates intramolecular toehold-mediated strand exchange, forming a stable intramolecular bivalent complex (**BI**) and displacing **O** in solution. Activation of **O** is monitored by a downstream reporter duplex (**Rep**), resulting in a stoichiometric increase in fluorescence. (**b**) Monitoring the release of output strand **O** as a function of toehold length $T$ in the absence (black) and presence of 5 nM of anti-HA antibody (red) or 5 nM anti-HIV1-p17 antibody (green). One normalized unit (n.u.) represents the fluorescence generated by the displacement of **Rep** by 1 nM **O**. (**c**) Apparent first-order rate constants ($k_{obs}$) obtained by fitting the kinetic traces shown in (**b**) using a single exponential. (**d**) Signal to background ratios as a function of $T$ obtained by dividing the antibody-templated first-order rate constants by the background rate constants. All experiments were performed with [**BO**] = 5.5 nM, [**I**] = 5 nM, [**Ab**] = 5 nM and [**Rep**] = 10 nM at 28 °C in TE/Mg$^{2+}$ buffer supplemented with 1 mg ml$^{-1}$ BSA. Error bars represent the standard error of estimated value of $k_{obs}$ calculated from the Fisher information matrix.

translator module, showing a maximal S/B ratio of ∼20 using a toehold of 3 nucleotides.

**ODE model of the ATSE module.** To provide a thorough understanding of the critical parameters that determine the performance of the ATSE reaction, we constructed a mechanistic kinetic model consisting of a set of ordinary differential equations (ODEs). The model describes the time-dependent concentrations of all start, intermediate and end products of the antibody-templated strand exchange reaction (Fig. 2a; Supplementary Methods). First, the rate constants that characterize the displacement of **Rep** by **O** ($k_{rep}$), and the binding of the peptide epitopes to the anti-HA antibody ($k_f$) were determined independently in separate experiments (Supplementary Figs 11 and 12). The rate constants of the background reaction ($k_{bg}$) as a function of toehold length were obtained by fitting of the fluorescence traces in Fig. 1b using the experimentally determined value of $k_{rep}$ as a fixed parameter and a model that describes the kinetics of the toehold-mediated strand exchange as a bimolecular reaction (Fig. 2b; Supplementary Table 5). The background

reaction was assumed to be effectively irreversible as DNA strand **O** is sequestered by an excess of reporter complex **F·Q**[18]. Finally, having established the kinetic parameters $k_f$, $k_{bg}$ and $k_{rep}$ and the $K_d$ for the antibody–epitope interaction allowed the determination of the rate constants ($k_{intra}$) for the antibody-templated intramolecular toehold-mediated strand exchange reaction as a function of toehold lenght by non-linear least square optimization of the ODE model to experimental data obtained in the presence of 2 nM anti-HA antibody (Fig. 2c; Supplementary Table 6). Using the experimentally obtained kinetic parameters $k_b$, $k_f$, $k_{rep}$, $k_{bg}$ and $k_{intra}$, we simulated the concentration of the individual reaction components as a function of time for the system with the optimal toehold length of 3 nucleotides (Fig. 2d; Supplementary Fig. 19). This simulation shows that the amount of free output **O** increases in the first few minutes of the reaction, indicating that the anti-HA antibody (**Ab**) rapidly induces the toehold-mediated strand exchange reaction to form the intramolecular cyclic complex (**Ab·B·I**). Subsequently, **O** reacts rapidly with **Rep** in the downstream displacement reaction, establishing a low pseudo steady-state concentration of **O** that is proportional to the concentration of

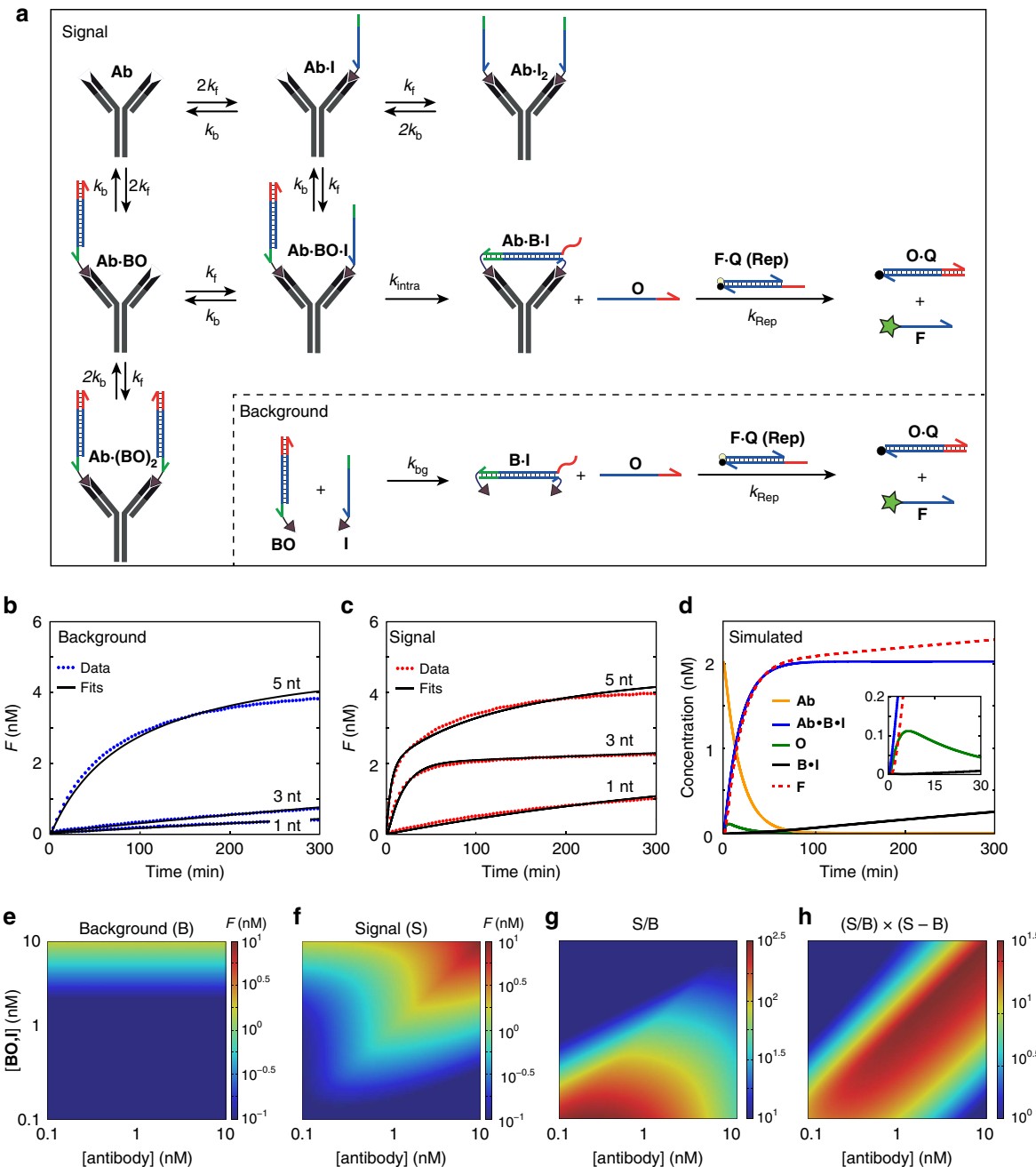

**Figure 2 | Non-linear least square analysis of data and simulations using a kinetic reaction model for ATSE.** (**a**) Scheme showing the reactions considered in the ATSE ODE model. (**b,c**) Non-linear least square optimization (solid black lines) of the ODE model to the experimental kinetic data for the background reaction (blue dots) and the anti-HA antibody-triggered reaction (red dots). (**d**) Kinetic speciation of free anti-HA antibody (**Ab**, orange), intramolecular cyclic complex (**Ab·B·I**, blue), free output strand (**O**, green), product of the background reaction (**B·I**, black) and the fluorescent product of the reporter duplex (**F**, dashed red) as obtained by simulations with the optimized parameters. Initial concentrations: [**Ab**] = 2 nM, [**BO**] = 5.5 nM, [**I**] = 5 nM, [**Rep**] = 10 nM. (**e**) Simulated background fluorescence as a function of initial [**BO, I**] and [**Ab**] after 3 h. (**f**) Simulated fluorescence of the antibody-templated toehold-mediated strand exchange reaction as a function of initial [**BO, I**] and [**Ab**] after 3 h. (**g**) Simulated signal-to-background (S/B) ratio as a function of initial [**BO, I**] and [**Ab**] after 3 h. (**h**) Simulated product of signal-to-background (S/B) ratio and dynamic range (S–B) as a function of initial [**BO, I**] and [**Ab**] after 3 h, yielding an empirical estimation of optimal initial conditions.

remaining free antibody. At this stage the increase in fluorescence closely mirrors the kinetics of the ATSE reaction as defined by the formation of the **Ab·B·I** complex. The analysis also shows that the background reaction only starts to contribute significantly to the fluorescence after 60 min, when the ATSE reaction has reached completion.

The ODE model can also be used to determine the optimal conditions for the ATSE reaction by assessing the influence of both the oligonucleotides (**BO**, **I**) and antibody (**Ab**) concentrations as well as the kinetics ($k_f$) and thermodynamics ($K_d$) of the antibody-peptide interaction. Figure 2e,f show the amount of **F** formed at $t = 180$ min for the background and antibody-

templated reaction, respectively. As expected, the background increases proportional to increasing [**BO**, **I**] and is independent of [**Ab**], whereas the antibody-templated signal strongly depends on both [**BO**, **I**] and [**Ab**]. The signal-to-background ratios (S/B, Fig. 2g), obtained by dividing the fluorescence formed in the presence of antibody (S) by the fluorescence of the background reaction (B), shows a maximum at [**Ab**] $\sim K_d$ and low concentrations of [**BO**, **I**]. The latter is caused by the strong, linear dependence of the background reaction on [**BO**, **I**]. However, since the absolute increase in fluorescence becomes increasingly difficult to distinguish above the background fluorescence for low concentrations of [**BO**, **I**], we defined an empirical formula for the optimal ATSE conditions as the product of the signal-to-background ratio (S/B) and the absolute dynamic range (S–B; Fig. 2h). The latter formula shows optimal performance at stoichiometric concentrations of antibody and oligonucleotides, but the ATSE reaction is relatively robust and antibody can

also be clearly detected above background at suboptimal stoichiometries of antibody and oligonucleotides.

**Antibody-triggered actuation of DNA-based logic circuits.** Having established ATSE as a robust and generic method to translate the presence of an antibody into a specific DNA sequence, we next explored the possibilities to use multiple antibodies as inputs for DNA-based molecular logic circuits. The anti-HA and anti-HIV1-p17 antibodies described above were used as two input antibodies and fluorescence was used to study the performance of various circuits. Figure 3a shows the construction of a circuit that allows multiplex detection of antibodies based on two parallel YES logic operators. In this system each antibody generates a specific DNA sequence, which can be detected simultaneously using two specific reporter duplexes, each generating a fluorescent output oligonucleotide with a

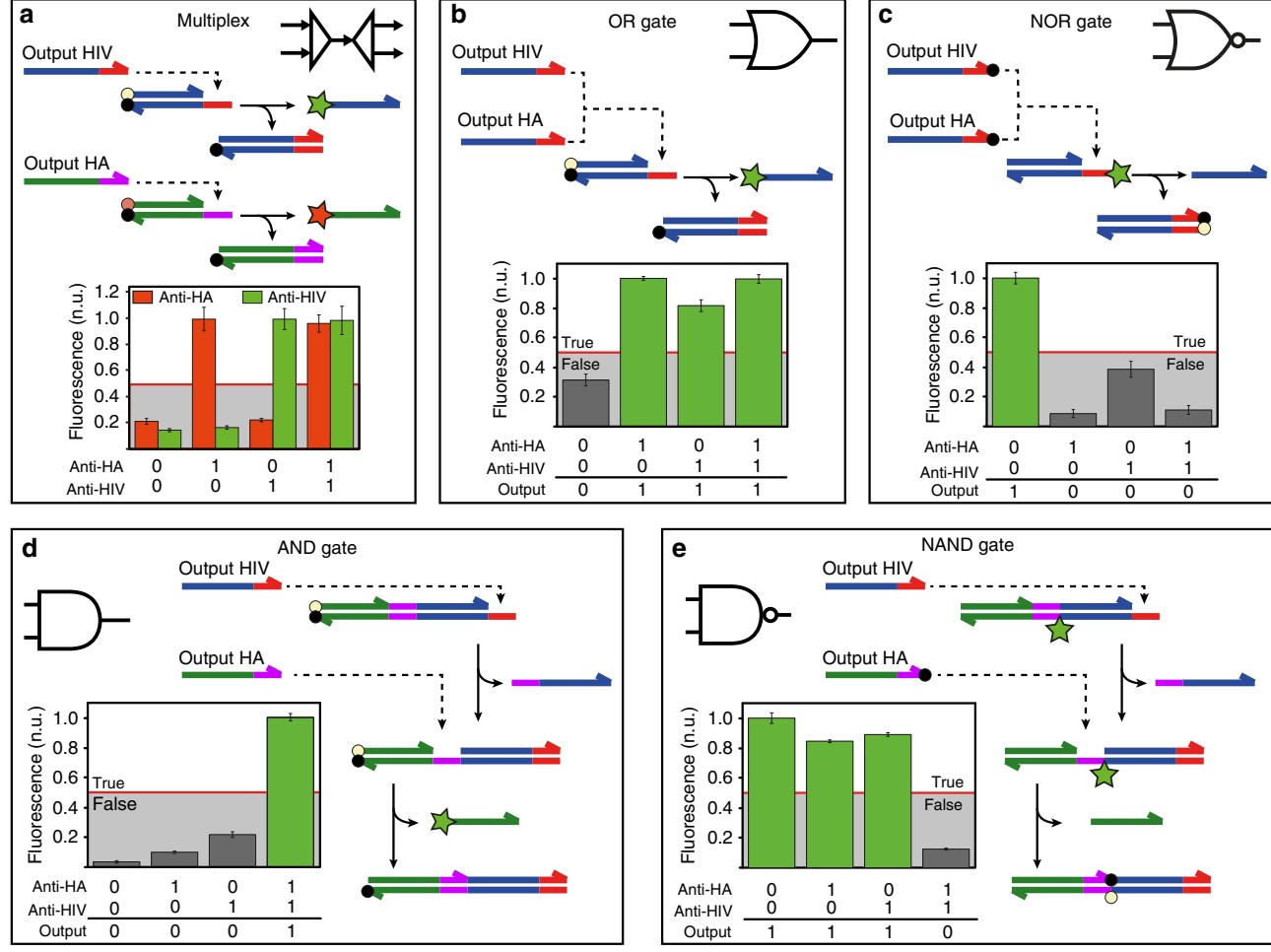

**Figure 3 | DNA-based molecular circuits allow complex signal processing in antibody detection.** (**a**) Multiplexed antibody detection. Two orthogonal translator modules process the anti-HA and anti-HIV antibodies into two unique ssDNA outputs. Each output is coupled to a downstream TMSD reaction to displace a different colour reporter duplex, allowing the identification of each antibody in a dedicated fluorescent channel (anti-HA = red, anti-HIV = green). (**b,c**) OR and NOR gates. Both input antibodies generate the same output sequence, which react with a single reporter duplex, resulting in increased (OR) or decreased (NOR) fluorescence when either one or both input antibodies are present. (**d**) AND gate. Sequential toehold exchange and toehold-mediated strand displacement reactions of the HIV and HA specific oligonucleotides displace a fluorescently labelled strand from the reporter complex. (**e**) NAND gate. Sequential toehold exchange and strand displacement reaction result in the quenching of an internally fluorescently modified reporter complex. Fluorescence levels were measured after 3 h incubation and corrected for background fluorescence measured at $t = 0$ h for the multiplex, OR and AND gates, or the fluorescence measured upon addition of an excess of output oligonucleotides for the NOR and NAND gates. Experiments were performed using [**BO**] = 5.5 nM, [**I**] = 5 nM, [**Ab**] = 10 nM and [**Gate/Rep**] = 1 nM at 28 °C in TE/Mg$^{2+}$ buffer supplemented with 1 mg ml$^{-1}$ BSA. Error bars represent the s.d. calculated from triplicate measurements.

different colour (red and green for anti-HA and anti-HIV1-p17, respectively). Figure 3a shows that there was indeed no cross-talk between the two detection pathways, as expected since each antibody was translated into a unique DNA sequence. Next we developed a complete set of Boolean logic operators by constructing OR, NOR, AND and NAND antibody detection circuits. OR and NOR logic operators (Fig. 3b,c) were constructed by translating both input antibodies to the same output sequence. In the OR gate, the output strand was detected using a single reporter duplex, showing a positive fluorescence signal in the presence of either or both of the input antibodies. The circuitry used for the NOR gate was very similar to that of the OR gate, but here quencher dyes were attached to the ATSE output strands, resulting in quenching of the fluorescence of the reporter duplex (Fig. 3c).

Figure 3d,e show the construction of AND and NAND gate antibody detection circuits. Here each antibody is translated into a different output DNA strand and sequential toehold-mediated strand exchange and displacement reactions on a reporter duplex are required to generate a fluorescence signal (AND gate) or quench a fluorescence signal (NAND gate). First, the output of the anti-HIV1-p17 translator module displaces a blocking strand from the reporter duplex, rendering a new toehold on the base strand. Via this newly formed toehold, the anti-HA output oligonucleotide can displace a fluorescent strand from the reporter duplex, yielding AND type logic behaviour. Figure 3d shows that a high level of fluorescence (1, true) was indeed only observed when both input antibodies were present. Using the same approach, a NAND gate can be constructed by functionalizing the anti-HA output oligonucleotide with a 3′-quencher and the reporter duplex with an internal fluorophore. This NAND circuit reported high fluorescence levels (1, true) in the presence of either one or none of the two antibodies, but low fluorescence when both antibodies were present. These examples show that coupling of the ATSE reaction with downstream DNA-based

molecular circuits allows the implementation of complex signal processing functionalities beyond those possible with current antibody detection approaches[35,36]. In general, antibody-based diagnostic assays based on multiple inputs will increase the specificity of current diagnostic approaches, especially for the detection of diseases that are typically characterized by a specific profile of antibodies such as autoimmune diseases[37,38].

**Autonomous sense-process-act molecular circuits.** Autonomous biomolecular systems that are able to obtain molecular information from their environment (sense), process this information and finally respond to their environment (act) represent attractive systems for the development of theranostic devices. Having established the ability to translate the presence of antibodies into specific DNA strands and subsequently use these as input for a variety of signal processing algorithms, we finally explored whether the DNA outputs could also be used for actuation. Here we explored two important classes of actuators, DNAzymes and DNA-directed control of enzyme activity, that can both be used for signal amplification. Our design of the DNAzyme actuator was based on the previously developed RNA-cleaving 8–17 DNAzyme (Fig. 4a)[39,40]. In the initial state the DNAzyme (**Dz**) is hybridized to an inhibitor oligonucleotide (**Inh**), rendering the DNAzyme catalytically inactive. A toehold on **Inh** allows **O**, the output oligonucleotide generated in the ATSE reaction, to bind and displace **Dz**. Subsequently, **Dz** can adopt its catalytically active conformation while binding with the substrate binding arms to the substrate (**S**). Cleavage of the RNA-nucleotide in **S** results in dissociation of the two product strands (**P₁** and **P₂**) from **Dz**, allowing it to bind to a new substrate oligonucleotide. Product formation was monitored by the introduction of FRET donor and acceptor fluorophores on each end of **S**. Figure 4a shows the conversion of the substrate **S** in the absence and presence of 5 nM anti-HA antibody. Following an initial lag phase

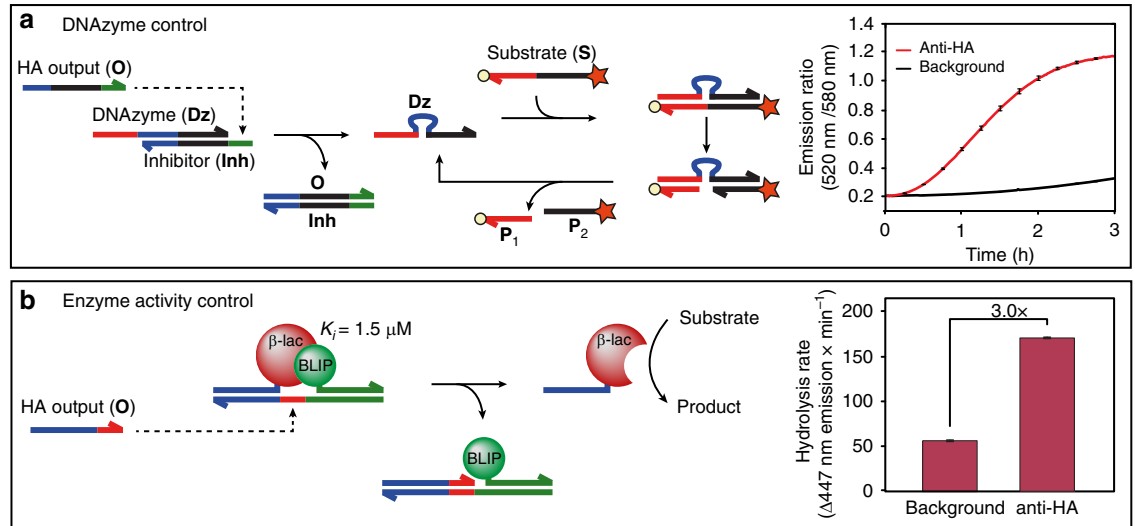

**Figure 4 | Antibody-Templated Strand Exchange allows antibodies to control DNAzyme and enzyme activity.** (**a**) Design and characterization of a DNAzyme-based actuator module. HA output **O**, generated by anti-HA antibody induced ATSE binds to a toehold on the inhibitor oligonucleotide (**Inh**) and displaces the DNAzyme (**Dz**), allowing it to adopt a catalytically active conformation while binding to substrate strand (**S**). After cleaving **S**, the product strands (**P₁** and **P₂**) spontaneously dissociate from **Dz** to complete the catalytic cycle. Experiments were performed with [**BO**] = [**I**] = [**Ab**] = [**DzInh**] = 5 nM, [**S**] = 10 nM at 28 °C in TE/Mg²⁺ buffer containing 1 mg ml⁻¹ BSA. (**b**) Design and characterization of a DNA-directed reporter enzyme actuator module. A weak binding enzyme-inhibitor pair, TEM1-β-lactamase and its inhibitor protein BLIP ($K_i$ = 1.5 µM), are each conjugated to an oligonucleotide, inducing their interaction upon hybridization to a shared template strand. The output oligonucleotide **O** generated in the anti-HA antibody induced ATSE reaction binds to an internal toehold on the template oligonucleotide and displaces the enzyme from the template strand, resulting in enzyme activation. Experiments were performed with [**BO**] = [**I**] = [**Ab**] = 5 nM, [**Reporter Enzyme**] = 1 nM at 28 °C in TE/Mg²⁺ buffer containing 1 mg ml⁻¹ BSA. Error bars represent the s.d. of triplicate measurements.

due to the upstream ATSE reaction, efficient activation of the DNAzyme is observed after 30–60 min, resulting in complete substrate conversion after 3 h. Although some substrate turn-over is eventually also observed in the absence of antibody, activity levels as estimated from Fig. 4a are at least 20-fold higher after 1 h incubation in the presence of antibody compared with the background reaction (Supplementary Fig. 6).

While DNAzymes by their nature are easily integrated into DNA molecular circuits, their catalytic diversity is limited compared to protein-based enzymes. We recently introduced a modular approach to control the activity of the reporter enzyme TEM1 β-lactamase by DNA-mediated modulation of the interaction of this enzyme with its inhibitor protein BLIP[41]. This system uses variants of TEM1-β-lactamase and BLIP that interact relatively weakly ($K_i = 1.5\,\mu M$)[42]. By conjugating each protein to an oligonucleotide their interaction can be induced via hybridization to a shared template strand, harbouring a toehold between the two duplexes (Fig. 4b). Via this toehold, the output oligonucleotide **O** can bind and displace the enzyme from the template strand, thereby disrupting the enzyme-inhibitor interaction and reactivating the enzyme. Using this system, a threefold higher β-lactamase activity was observed in the presence of the anti-HA antibody after 3 h incubation compared with the background reaction in the absence of antibody. These examples demonstrate how the modular nature of oligonucleotide hybridization can be used as a universal 'programming language' to connect antibody activity to in principle any DNA-controlled molecular process, including the regulation of protein-protein interactions.

## Discussion

We have shown that Antibody-Templated Strand Exchange (ATSE) of peptide-functionalized DNA strands provides a unique and robust molecular approach to translate the presence of an antibody into a ssDNA output. Both thermodynamic and kinetic effects contribute to the remarkable efficiency of ATSE. First, the bivalent peptide-dsDNA product of the ATSE reaction forms a highly stable 1:1 cyclic complex with its bivalent target antibody, thus making the displacement reaction thermodynamically favourable[33,34]. Second, colocalization of the peptide-functionalized oligonucleotides on the two antigen binding domains increases their effective concentration, hence enhancing the rate of the exchange reaction. An important application of the ATSE reaction is that it allows the use of DNA-based molecular circuits in antibody-based diagnostics, introducing complex signal-processing capabilities beyond those achievable in convential immunoassays. In addition to the logic gates and multiplex detection demonstrated in this work, many other features of DNA-based molecular circuits could be employed, including tresholding, signal amplification, feedback and signal modulation[9,43,44]. The importance of ATSE to the field of DNA-nanotechnology is that it provides a generic method to use antibodies as inputs for DNA-based molecular computing and the actuation of 3D DNA-nanoarchitectures. Since antibodies can be generated that bind with high affinity and specificity to almost any molecular target, any of these biomarkers can now also be considered as potential inputs for DNA-based molecular circuits, by competing with ATSE-mediated generation of DNA input strands. As a generic mechanism that allows protein-based control of DNA circuits, ATSE complements previously developed molecular approaches for DNA-based control of protein activity. The development of these and other molecular strategies to integrate the rich functional properties of antibodies and other proteins with the inherent programmability of DNA-nanotechnology will provide access to truly autonomous biomolecular systems with sophisticated signal integration, processing and actuation properties.

## Methods

**Synthesis and purification of peptide oligonucleotide conjugates.** Peptide epitopes were synthesized using automated standard Fmoc peptide synthesis on Rink amide MBHA resin. Peptides were cleaved from the resin using a mixture of TFA/TIS/H$_2$O/EDT (92.5:2.5:2.5:2.5% v/v) for 3 h at room temperature and precipitated in ether at $-30\,^\circ C$. Purification was performed using reversed phase HPLC-MS on a Shimadzu LC-8A HPLC system with a VYDAC protein & peptide C18 column. Purity and correct mass were confirmed by LCMS on an Applied Biosystems Single Quadrupole ESI API-150EX in positive mode. All oligonucleotides were purchased HPLC-purified from Integrated DNA Technologies. Amino-modified oligonucleotides were dissolved in PBS pH 7.2 and mixed with 20 equivalents of Sulfo-SMCC (Thermo Fisher Scientific) in 50% DMSO for 2 h at room temperature. The oligonucleotides were isolated using ethanol precipitation and subsequently dissolved in 100 mM sodium phosphate pH 7 and mixed with 10 equivalents of the peptide epitope for 2 h at room temperature. Finally, the obtained conjugates were purified by RP-HPLC on a GraceAlpha C18 (250 × 4.6 mm) column using a gradient of 5–50% acetonitrile in 100 mM TEAA pH 7.0. Correct masses of the POCs were confirmed by FIA ESI ion-trap mass spectrometry in negative mode.

**Duplex preparation.** All oligonucleotides and POCs were dissolved in TE buffer (10 mM Tris-HCl, 1 mM EDTA, pH 8.0) and diluted to a concentration of 50 μM. Base-Output duplexes were obtained by mixing the Base and Output strands in TE/Mg$^{2+}$ to a final concentration of 2.5 μM and allowed to hybridize for 1 h at room temperature. Reporter duplexes were obtained by mixing the individual oligonucleotides in TE/Mg$^{2+}$ to a final concentration of 5 μM and annealed from 90 to 10 °C in 1 h. AND and NAND gate reporter duplexes were further purified by 10% native PAGE.

**Fluorescence measurements.** For the screening of optimal toehold length **BO** (5.5 nM), antibody (5 nM) and Reporter duplex (**Rep**, 10 nM) were mixed in TE/Mg$^{2+}$ supplemented with 1 mg ml$^{-1}$ BSA to a total volume of 190 μl and allowed to equilibrate for 1 h at 28 °C. Finally, the ATSE reaction was induced by the addition of **I** (5 nM) to a final volume of 200 μl and fluorescence intensities were recorded with a platereader for 3 h at 28 °C. Obtained fluorescence intensities were normalized to a negative control containing no **BO**, and a positive control where **BO** was substituted with 5.5 nM free **O**. This results in 1 normalized unit (n.u.) to correspond to fluorescence generated by 1 nM of **O**. Assuming stoichiometric translation of antibody to output oligonucleotide 1 n.u. therefore corresponds to the fluorescence intensity generated by 1 nM of antibody. For downstream multiplexing and logic operations **BO** (5.5 nM) and **I** (5 nM) for both the anti-HA and anti-HIV1-p17 translator modules were mixed with **Rep** (1 nM) in TE/Mg$^{2+}$ supplemented with 1 mg ml$^{-1}$ BSA in a final volume of 180 μl. To initiate the ATSE reactions, 10 μl of either antibody or buffer was added, yielding samples containing none, one or both antibodies at a final concentration of 10 nM.

**Data availability.** The data that support the findings of this study are available from the corresponding author upon reasonable request.

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

## Acknowledgements

This work was supported by European Research Council (ERC) Starting Grants (280255 SwitchProteinSwitch and 677313 BioCircuit), an ECHO-STIP grant from the Netherlands Organization for Scientific Research (NWO, 717.013.001) and funding from the Ministry of Education, Culture and Science (Gravity programme, 024.001.035).

## Author contributions

W.E., L.H.H.M. and B.S. performed experiments, T.F.A.d.G. and M.M. supervised the research and provided advise, W.E., L.H.H.M., T.F.A.d.G. and M.M. analysed the data and wrote the manuscript.

## Additional information

**Competing financial interests:** The authors declare no competing financial interests.

**Publisher's note**: 

