## [Peer Review File · Nature Communications]

Reviewer #1 (Remarks to the Author):

This manuscript introduces a way to couple antibody to DNA strand displacement systems. This is an important topic. Immunochemistry is bound to implement better and smarter testing strategies, and therefore needs to explore new coupling strategies with various molecular reporters or devices.

The manuscript is based on the large body of work that has been done to develop DNA-based means of performing logic operations and information processing in vitro. A logical extension for the body of work is to harness it in diagnostic applications. One question is to find out the most efficient way to bridge the biomolecular target (the antibody here) to DNA. The authors propose to use the divalency of the antibody structure to put DNA gates in close contact, effectively increasing the rate of a strand exchange reaction. Once the antibody signal is concerted in a single stranded DNA strand, various designs are demonstrated for downstream processing and reporting. The approach given by the authors is sound, the experiment clean and well presented, with the only reservation that the limit of detection is not discussed in the context of clinical relevance. Also this particular design is not the only possibility and it could be discussed and compared to other proposals from the literature.

Overall, I recommend the publication of this manuscript, after the few minor comments above and below are addressed.

Fig1: what is n.u. ?

The explanation about why the background reaction in Fig 2 is drawn as irreversible (whereas it is described in the text as thermodynamically unfavorable) is given only in SI S19 point 5. This is a bit late.

Fig2b, the number written in the plots are not explained. They probably represent toehold length, but this should be stated somewhere. Also, is this data the same as presented in Fig1 (but converted to nM)? It is not clear.

In Fig2e, it is a bit weird to plot the background intensity as a function of antibody, since this reaction is by definition the one that does NOT depend on [antibody]. Also, the use of a linear scale in these heat maps flattens the plots and as a consequence, the shape of the S/B plot cannot be guessed from the background and signal plots. A log scale would probably fix that.

The authors do not really address the limit of detection of their method. (even if it is possibly not the most relevant point here, as the purpose is more to show the possibility to link antibody chemistry to DNA Strand displacement circuits). However, when they introduce downstream reporting strategy (with additional amplification modules introduced in Fig4), the authors could compare the LOD with the direct circuit presented in the beginning of the paper. Since therapeutics are used as a motivation, it would be interesting to discuss the relevance of this LOD to clinical applications.

The presentation of DNA encoded "molecular circuitry" (and associated references) could be better balanced and not overly focused on strand displacement cascades. A couple of references to enzymatic approaches could help the uninformed reader to obtain an informative picture of the field, especially since enzymatic circuits typically provide the amplification effect that could be required here. This is also important since the paper later uses an elegant way to control enzymatic activity with DNA strands.

While the rate acceleration of 20~100 described in the paper is fine, one can only wonder if it could be pushed further, and how. The only design element that has been optimized is the length of the toehold, but how the total length of BO was selected is not explained. Is the current choice better than a slightly longer, or slightly shorter duplex? This may be relevant since the antibody imposes geometric constraints on the intramolecular reaction.

Typos:

- separate
- circuitery

Reviewer #2 (Remarks to the Author):

In the current work, the authors introduced Antibody Templated Strand Exchange (ATSE) in constructing molecular circuits. The use of the characteristic bivalent Y-shaped architecture of antibodies is certainly powerful, by promoting DNA strand exchange both thermodynamically and kinetically. However, ATSE is not as generic as the authors claimed in this manuscript. The intracellular application (e.g. smart drug delivery and even the rewiring of natural signalling pathways *in vivo* based on DNA circuits) of this antibody-based approach is very likely impossible. The conjugation and purification of DNA-peptide complex is quite laborious. Peptides without cysteine residue can not be conjugated to DNA-NH₂ according to the protocol in Supplementary Information. Furthermore, after conjugation with DNA, binding affinity of peptides to antibodies may be dramatically disturbed. Therefore, this manuscript is an interesting extension/continuation of work that has been previously done by authors. However, it does not have sufficient novelty to warrant publication in Nature Communications.

The experiments are well-designed and the presentation is clear. The following specific points should be properly addressed:

[1] Authors determined dissociation constant (K_d) for fluorescein labeled HA binding to the antibody (Figure S8, P4, line80) and the value of K_d was calculated to be 0.2 nM. In ASTE process, it is DNA-peptide binding to antibody not fluorescein-peptide. How is the K_d for fluorescein-DNA labeled HA to the antibody? Does DNA attachment significantly influence the binding affinity of HA to antibody?

[2] This question refers to Question 1. Authors extended ASTE approach to anti-HIV1-p17 antibody. The S/B ratio for anti-HIV1-p17 antibody was ~ 20 , compared to ~ 100 for anti-HA antibody (Figure 1d). The reason for the lower ATSE rate was proposed to be the lower affinity of HIV1-p17 to antibody, with a K_d of 42 nM based on literature. It is better to calculate the K_d of DNA-HIV1-p17 to antibody and then compare the two K_d values of DNA-peptide binding to antibody. Moreover, it will be great to include extra data of a DNA-peptide/antibody system with a K_d between HA and HIV1-p17 systems.

[3] Since the background reaction only starts to contribute significantly to the observed fluorescence after 60min, when the ATSE reaction has reached completion (P7, line 145), why the analysis of Figure 2e and 2f was based on data at 180 min not 60 min? Is there a time dependency of the estimation of optimal initial conditions (Figure 2g and 2h)?

[4] This question refers to Question 3. In Figure 3, the fluorescence levels of those DNA circuits was measured after 3 h. How about the results after 1 h?

[5] Authors claimed that "activity levels as estimated from Fig 4a are at least 20-fold higher after 1 h in cubation in the presebnce of antibody compared to the background reaction". How did they get this conclusion?

Minor points:

[6] To make this manuscript easier to read, it is better to point out which system (HA or HIV1-p17) is employed in text, Figure 2 and relevant supporting figures.

[7] In Figure 4a, the pentagram of P1 and the spherosome of P2 should be switched.

Reviewer #3 (Remarks to the Author):

The manuscript describes the "antibody templated strand exchange" or ATSE, which specifically translates the presence of antibodies to unique DNA strand outputs. Subsequently, ATSE makes monoclonal antibodies (and potentially their specific targets) compatible with the well established molecular computing systems powered by nucleic acids. The design is elegant and very smart. The reaction mechanisms is carefully studied by experiments and modeling. Finally, the ATSE reactions are coupled with a few simple logic gates and enzyme catalyzed signal-amplification reactions to fully prove the concept.

The study is quite thorough and most claims are well supported by substantial amount of data and careful analyses. Given the interest of molecular computing in nanotechnology and synthetic biology communities, this work should be interesting to a broad range of readers.

Nevertheless I have the following concerns/questions that I want the authors to address:

(1) According to main text, Figure 1b shows the ATSE with toehold length increasing from 1nt to 6nt. However, the DNA sequences shown in Table S1 says otherwise. BxOxO has 3nt overhang (TAG) for Ix to bind. So while Figure 1b shows $T=1,2,3\dots 6$, the toehold length should actually be 4,5,6...9. Otherwise the data do not make sense: for example it is thermodynamically highly unfavorable for the background reaction to take place when toehold is 4 nt because it would result in an higher enthalpy state (i.e. less base pair formed). This should be clarified in the main text.

(2) It is mentioned multiple times in the manuscript that ATSE works because of a. the formation of a stable cyclic structure that involves bivalent interactions and b. higher local concentration of reactants or faster intramolecular interactions. I think this explanation is superficial and misleading to say the least. To me, it works because of one reason: entropy gain, i.e. the release of the output strand from the antibody-DNA complex. Consider the following thought experiment: an "antibody" that has one additional binding pocket for "O" so that it would still be part of the complex and stay nearby after being displaced from "B". The strand displacement will never happen at appreciable level if BO forms more base pairs than BI because "O" can easily come back to kick "I" off "B". Am I wrong?

(3) I strongly suggest replacing Figure 2a with Figure S7, which shows the formation of Ab(I2) and Ab(BO)2. Otherwise readers can be left wondering why those complexes are not included in the model (but in fact they were).

(4) It would be nice to show a zoom-in view of the first ~30 minutes of Figure 2d. As presented it is hard to see the level of "O" (green curve).

(5) Can authors provide explanations as to why the signal-to-background ratio in Figure 3b (OR gate, roughly 3.5:1) and Figure 4b (enzyme activity control, roughly 3:1) are much worse than what's shown in Figure 1 (at least 20:1, as high as 100:1)?

(6) I find the sentence that starts at line 347 and ends at line 349 very hard to understand. Exactly what is considered 0 and 1 fluorescent unit? The explanation provided in SI Section 2a seems much clearer to me. My personal preference of course does not matter but I think breaking the sentence in question into two, one explaining negative control (0) and another explaining positive control (1), may help reader like me.

(7) Typo: line 134, "lenght" should be "length".

Department of Biomedical Engineering

Molecular Science and Technology
Laboratory of Chemical Biology

Den Dolech 2, 5612 AZ Eindhoven
P.O. Box 513, 5600 MB Eindhoven
The Netherlands
Internal address: Helix STO 3.22
www.tue.nl

Dr Chiara Pastore
Associate Editor Nature Communications

Date

November 2nd 2016

Contact

Prof. Dr. M. Merkx
T +31 40 24724728
m.merkx@tue.nl

Dear Dr Pastore,

Thank you for sending us the reviewers' assessment of our manuscript. We are pleased that all reviewers appreciated our work and have used their feedback to further improve our manuscript. Please find a point-by-point response below.

Reviewer 1

[1] Fig 1: what is n.u.?

In figure 1 the obtained fluorescence values were normalized to controls that either lack BO or are spiked with 5.5 nM of free output. N.u. stands for normalized unit, which corresponds to the fluorescence that is generated when 1 nM output displaces the reporter duplex. We have added the following sentence to the legend of Figure 1 for clarification "One normalized unit (n.u.) represents the fluorescence generated by the displacement of Rep by 1 nM O."

[2] The explanation about why the background reaction if fig 2 is drawn as irreversible (whereas it is described in the text as thermodynamically unfavorable) is given only in SI S19 point 5. This is a bit late.

In principle the reaction is reversible, but since any product O that is formed is quickly trapped by reaction with the reporter duplex it is effectively irreversible and modelled accordingly. As suggested, we have moved the explanation from the SI to the main text. ("The background reaction was assumed to be effectively irreversible as DNA strand O is sequestered by an excess of reporter complex F·Q.")

[3] Fig2b, the number written in the plots are not explained. They probably represent toehold length, but this should be stated somewhere. Also, is this data the same as presented in Fig1 (but converted to nM)? It is not clear.

The numbers in Fig2b are indeed toehold lengths. We have adjusted Figure 2b and 2c to make this clear.

The data in Fig2c is not the same as presented in Fig1b. In Fig2c a lower antibody concentration (2 nM) was used. Using a lower concentration of antibody compared to the concentrations of BO (5.5 nM) and I (5 nM) is advantageous for fitting the kinetic traces. E.g. in the experiment using a toehold of 3 nucleotides, both the ATSE reaction (which dominates in the first 60 min) and the background reaction (after the ATSE has reached completion) can be modeled accurately. To avoid any confusion, we now also mention in the main text that an antibody concentration of 2 nM was used. ("optimization of the ODE model to experimental data obtained in the presence of 2 nM anti-HA antibody (Fig. 2c and Table S6).")

[4] In Fig2e, it is a bit weird to plot the background intensity as a function of antibody, since this reaction is by definition the one that does NOT depend on [antibody]. Also, the use of a linear scale in these heat maps flattens the plots and as a consequence, the shape

Date
November 2nd, 2016

Page
2 From 6

of the S/B plot cannot be guessed from the background and signal plots. A log scale would probably fix that.

We agree that using log scales makes it easier to interpret these plots and have adjusted Figures 2e- 2h accordingly. The background reaction is indeed not dependent on antibody concentration. However, to allow easy comparison of 2e – 2h we plotted all of them in a similar manner as heat maps. In this way it is easier to see how plots 2g and 2h are derived from plots 2e and 2f.

[5] The author do not really adress the limit of detection of their method. (even if it is possibly not the most relevant point here, as the purpose is more to show the possibility to link antibody chemistry to DNA Strand displacement circuits). However, when they introduce downstream reporting strategy (with additional aplification modules introduced in fig4), the authors could compare the LOD with the direct circuit presented in the beginning of the paper. Since theranostics are used as a motivation, it would be interesting to discuss the relevance of this LOD to clinical applications.

Reaching a low LOD was indeed not the prime motivation for this study. Nonetheless, we agree that it may be useful to provide an LOD of the direct circuit in order to compare our method with other analytical methods. We therefore determined the LOD for the ATSE reaction with direct detection using the reporter duplex as described in Figure 1, yielding an LOD of 23 pM. A figure showing the LOD determination has been added to the SI (Figure S5), and we mention the LOD in the main text. (“The limit of detection (LOD) was determined to be 23 pM under these conditions (Fig. S5.”). Since the circuits presented in figure 4 were not optimized for LOD, a comparison of the LODs of the downstream systems to that of the direct circuit is not very useful. E.g. the β -lactamase-BLIP system has a relatively large S/N because the enzyme is not fully inhibited in the absence of any trigger strand.

[6] The presentation of DNA encoded “molecular circuitry” (and associated references) could be better balanced and not overly focused on strand displacement cascades. A couple of reference to enzymatic approaches could help the uninformed reader to obtain an informative picture of the field, especially since enzymatic circuit typically provide the amplification effect that could be required here. This is also important since the paper later uses an elegant way to control enzymatic activity with DNA strands

We agree that enzymatically controlled DNA circuits are another important class of molecular circuits. We now explicitly mention this in the introduction, including references to representative papers. (“DNA-based molecular circuits capable of adaptation, oscillations and bistability have been created using enzymes involved in DNA synthesis and degradation such as DNA polymerases, nickases and exonucleases. [Kim, J., Khetarpal, I., Sen, S. & Murray, R. M. Nucl. Acids Res. 2014], [Montagne, K., Plasson, R., Sakai, Y., Fujii, T. & Rondelez, Y. Molecular Systems Biology. 2011], [Padirac, A., Fujii, T. & Rondelez, Y. PNAS. 2012]”)

[7] While the rate acceleration of 20~100 described in tge paper is fine, one can only wonder if it could be pushed further, and how. The only design element that has been optimized is the length of the toehold, but how the total length of BO was selected is not explained. Is the current choice better than a slightly longer, or slightly shorter duplex? This may be relevant since the antibody imposes geometric constraints on the intramolecular reaction.

We realize that we did not explicitly discuss our rationale for choosing a duplex length of 24 nucleotides. Antibodies can accommodate different lengths of the duplexes due to a certain flexibility in the hinge-region of the antibody. In our previous work we found that bivalent peptide-dsDNA constructs with duplex lengths of 20 and 35 nucleotides bind equally well and both form tight 1:1 complexes, whereas linkers of 50 nucleotides also

Date
November 2nd, 2016

Page
3 From 6

form 2:2 complexes. We have added a short discussion of this important design consideration to the beginning of the Results section where we introduce our experimental system, including references to our previous work. (“The length of the strands was chosen such that the bivalent peptide-dsDNA product **BI** contained 24 base pairs, a length that we have previously shown to be optimal for bivalent binding to the target antibody.^{33,34c}”).

Reviewer 2

General:

ATSE is not as generic as the authors claimed in this manuscript. The intracellular application (e.g. smart drug delivery and even the rewiring of natural signalling pathways in vivo based on DNA circuits) of this antibody-based approach is very likely impossible. The conjugation and purification of DNA-peptide complex is quite laborious. Peptides without cysteine residue can not be conjugated to DNA-NH2 according to the protocol in Supplementary Information. Furthermore, after conjugation with DNA, binding affinity of peptides to antibodies may be dramatically disturbed.

We agree that intracellular applications of ATSE are at present nearly impossible, but we also did not make any claims in our manuscript in this direction. While the preparation of peptide-DNA conjugates indeed requires careful HPLC purification, conjugation efficiencies in our hands are 80-100%. In addition, commercial suppliers have started to offer peptide-DNA conjugates (BioSynthesis, LifeTein). The remark that peptides without a cysteine cannot be conjugated is peculiar, since in those cases one can add a cysteine at the N-or C-terminus of the peptide, as we did for both the HA and HIV-p17 epitopes. In fact it is more challenging when the original peptide epitope contains one or more native cysteines, e.g. in disulfide-containing cyclic peptides. In those cases, alternative bioconjugation methods are available, however, such as strain-promoted click chemistry. While conjugation of the peptide to DNA can affect the binding affinity, we have verified that this is not the case for the two examples shown in this work (see below).

[1] Authors determined dissociation constant (K_d) for fluorescein labeled HA binding to the antibody (Figure S8, P4, line80) and the value of K_d was calculated to be 0.2 nM. In ASTE process, it is DNA-peptide binding to antibody not fluorescein-peptide. How is the K_d for fluorescein-DNA labeled HA to the antibody? Does DNA attachment significantly influence the binding affinity of HA to antibody?

The fluorescently labeled HA peptide was used for determination of the K_d , because we also used this probe to determine kinetic parameters using stopped-flow fluorescence anisotropy. The latter experiments could not be done using fluorescently labeled peptide-DNA conjugates, because the increase in fluorescence anisotropy was too small to reliably detect using our stopped-flow machine. However, we agree with the referee that conjugation to DNA could affect the binding affinity of the peptide to the antibody. We therefore carefully repeated the antibody binding assays with a fluorescently labeled peptide-DNA conjugate. Similar K_d values of 0.27 and 0.78 nM were obtained for the fluorescently labeled peptide and fluorescently labeled peptide-DNA conjugates, respectively. These additional titration experiments have been added to the supporting information (Figure S9).

[2] This question refers to Question 1. Authors extended ASTE approach to anti-HIV1-p17 antibody. The S/B ratio for anti-HIV1-p17 antibody was ~20, compared to ~100 for anti-HA antibody (Figure 1d). The reason for the lower ATSE rate was proposed to be the lower affinity of HIV1-p17 to antibody, with a K_d of 42 nM based on literature. It is better to calculate the K_d of DNA-HIV1-p17 to antibody and then compare the two K_d values of

Date
November 2nd, 2016

Page
4 From 6

DNA-peptide binding to antibody. Moreover, it will be great to include extra data of a DNA-peptide/antibody system with a K_d between HA and HIV1-p17 systems.

As suggested by the referee, we also repeated these binding experiments both for the peptide epitope and the peptide-DNA conjugates. Again, similar K_d values of 21 and 16 nM were obtained for the peptide and peptide-DNA conjugates, respectively. These additional titration experiments have been added to the supporting information (Figure S10). We have adjusted the K_d values mentioned in the text to the values obtained for the peptide-DNA conjugates.

[3] Since the background reaction only starts to contribute significantly to the observed fluorescence after 60min, when the ATSE reaction has reached completion (P7, line 145), why the analysis of Figure 2e and 2f was based on data at 180 min not 60 min? Is there a time dependency of the estimation of optimal initial conditions (Figure 2g and 2h)?

The ATSE reaction modelled in Fig2d indeed reaches completion after 60 minutes, but this was for a relatively high antibody concentration and an optimal toehold length. We chose to show the data after 180 min because the ATSE will be slower at lower concentrations of antibody and/or BO and I (modelled in Fig2e and f). The time point of 180 min was chosen to best accommodate these diverse reaction conditions. In fact, time is one of the parameters that determine the optimal signal-to-background and similar plots could be generated showing the various parameters as a function of e.g. time and BO/I concentration for a fixed antibody concentration, or time and antibody concentration for a fixed concentration of BO/I.

[4] This question refers to Question 3. In Figure 3, the fluorescence levels of those DNA circuits was measured after 3 h. How about the results after 1 h?

Since the AND and NAND gates are based on two sequential toehold-mediated strand displacement reactions (three if you include the ATSE reaction itself) the rates of these reactions are slower than the rates of the single toehold-mediated strand exchange reactions (the OR and NOR gates) and require 3h to reach completion. To keep the analysis of the logic circuits consistent, we report the fluorescence intensities of all gates after 180 minutes.

[5] Authors claimed that “activity levels as estimated from Fig 4a are at least 20-fold higher after 1 h in cubation in the presence of antibody compared to the background reaction”. How did they get this conclusion?

The activity levels in absence and presence of antibody were determined by deriving the slope of the emission ratio at 1 hour incubation (i.e. the change in emission ratio as a function of time). This slope is a measure for the activity of the DNAzyme. We have included a figure in the supporting information (Figure S6) showing our data analysis and have included a reference to this figure in the main text.

[6] To make this manuscript easier to read, it is better to point out which system (HA or HIV1-p17) is employed in text, Figure 2 and relevant supporting figures.

We agree. We have adjusted the legends of figures 2 and 4 and all relevant figures in the SI accordingly. We have also added this information at relevant places in the main text.

[7] In Figure 4a, the pentagram of P1 and the spherosome of P2 should be switched.

We thank the reviewer for catching this mistake, we have switched the pentagram and spherosome as suggested.

Date
November 2nd, 2016

Page
5 From 6

Reviewer 3

[1] According to main text, Figure 1b shows the ATSE with toehold length increasing from 1nt to 6nt. However, the DNA sequences shown in Table S1 says otherwise. B_xO_x0 has 3nt overhang (TAG) for I_x to bind. So while Figure 1b shows $T=1,2,3...6$, the toehold length should actually be 4,5,6...9. Otherwise the data do not make sense: for example it is thermodynamically highly unfavorable for the background reaction to take place when toehold is 4 nt because it would result in an higher enthalpy state (i.e. less base pair formed). This should be clarified in the main text.

The confusion may have been caused by the fact that we colored the TAG sequence in the original Table S1 purple to highlight that this sequence represents the toehold in the optimal system with the 3 nt overhang. To avoid this confusion we have removed this color coding in the revision. The reviewer is not correct in thinking that toehold lengths run from 3-9 nt, they really vary between 0 and 6 as shown in Figure 1b and as discussed in the text. Inspection of table S1 also shows this, e.g. hybridization of B_x with O_{x0} shows that they form a duplex with a toehold length of 0 (not 3). The reviewer is correct that the displacement reaction between I_x and B_xO_{x4} (toehold length of 4) is thermodynamically unfavorable. The fact that this background reaction is still observed is because it is coupled to a downstream displacement reaction (reaction of O_{x4} with the reporter duplex) that is thermodynamically favorable (6 additional base pairs are formed) and kinetically irreversible.

[2] It is mentioned multiple times in the manuscript that ATSE works because of a. the formation of a stable cyclic structure that involves bivalent interactions and b. higher local concentration of reactants or faster intramolecular interactions. I think this explanation is superficial and misleading to say the least. To me, it works because of one reason: entropy gain, i.e. the release of the output strand from the antibody-DNA complex. Consider the following thought experiment: an "antibody" that has one additional binding pocket for "O" so that it would still be part of the complex and stay nearby after being displaced from "B". The strand displacement will never happen at appreciable level if BO forms more base pairs than BI because "O" can easily come back to kick "I" off "B". Am I wrong?

The reviewer's remarks made us realize that we didn't explain very well why the formation of a stable cyclic structure between the bivalent antibody and the bivalent reaction product provides an important thermodynamic driving force for the ATSE reaction. In previous work we have shown that binding of a bivalent peptide-dsDNA ligand to the anti-HIV1p17 antibody is 500-fold stronger than the monovalent interaction. For the anti-HA antibody this difference between mono- and bivalent binding is even larger. This large gain in affinity (or better avidity) upon formation of the bivalent ligand results from the rigid nature of the dsDNA linker, and also effectively blocks the reverse reaction. We have added this explanation to the first paragraph of the Results section where we introduce the principle of the ATSE reaction, including references to the previous work. ("We recently showed that bivalent peptide-dsDNA ligands form very tight 1:1 complexes with their target antibody, showing a 500-fold increase in affinity compared to the monovalent peptide-antibody interaction.^{33,34}")

[3] I strongly suggest replacing Figure 2a with Figure S7, which shows the formation of $Ab(I_2)$ and $Ab(BO)_2$. Otherwise readers can be left wondering why those complexes are not included in the model (but in fact they were).

We agree and have extended the thermodynamic scheme shown in Fig 2a to include the formation of $Ab(I_2)$ and $Ab(BO)_2$.

Date
November 2nd, 2016

Page
6 From 6

[4] It would be nice to show a zoom-in view of the first ~30 minutes of Figure 2d. As presented it is hard to see the level of "O" (green curve).

As suggested by the reviewer, we have added a zoom-in-view of the first 30 minutes of the reaction that more clearly shows the level of "O".

[5] Can authors provide explanations as to why the signal-to-background ratio in Figure 3b (OR gate, roughly 3.5:1) and Figure 4b (enzyme activity control, roughly 3:1) are much worse than what's shown in Figure 1 (at least 20:1, as high as 100:1)?

The lower signal-to-background ratio for the OR gate can be explained by the fact that Figure 3a shows the fluorescence after 3 hours incubation, while Figure 1 shows the ratios of the rates (k_{obs}). In addition, for the OR gate two-fold higher oligonucleotide concentrations were used in the ATSE reaction (for each target antibody), which increases the rate of background reaction of the ATSE reaction.

The lower signal-to-background ratio in Figure 4b results primarily from the substantial amount of background enzymatic activity that is present even in the absence of any HA output strand. This background activity results from the non-complete inhibition by the inhibitor protein BLIP in this system.

[6] I find the sentence that starts at line 347 and ends at line 349 very hard to understand. Exactly what is considered 0 and 1 fluorescent unit? The explanation provided in SI Section 2a seems much clearer to me. My personal preference of course does not matter but I think breaking the sentence in question into two, one explaining negative control (0) and another explaining positive control (1), may help reader like me.

To make the explanation of normalizing the experimental data to a positive and negative control clearer we have changed this sentence as suggested by the reviewer. ("Obtained fluorescence intensities were normalized to a negative control containing no **BO**, and a positive control where **BO** was substituted with 5.5 nM free **O**. This results in 1 normalized unit (n.u.) to correspond to fluorescence generated by 1 nM of **O**. Assuming stoichiometric translation of antibody to output oligonucleotide 1 n.u. therefore corresponds to the fluorescence intensity generated by 1 nM of antibody.").

We hope that with these improvements the manuscript can now be accepted for publication. Please let me know if you require further information.

Yours sincerely,

Maarten Merckx

Reviewer #1 (Remarks to the Author):

I think the authors have made a good job and an honest response to the first round of review, and answered satisfyingly to my comments. I have also checked the remarks from the other referees and the corresponding rebuttal and found no specific technical issue.

This paper makes an interesting contribution to the field and provides a nice approach to connecting DNA devices and antibody biomarkers. It is now well balanced in the discussion part. As such I am OK for publication in the current form.

Reviewer #2 (Remarks to the Author):

I have read the responses from the authors. They have satisfactorily responded to my concerns. And the manuscript has been significantly improved. I think this version has been appropriate for publication in Nature Communications.

Reviewer #3 (Remarks to the Author):

I saw that I misinterpreted part of the data and thank the authors for clarifying that. In addition, authors have adequately addressed all my concerns. I continue to support this manuscript for publication in Nat Commun.